# Frontline and Relapsed Rhabdomyosarcoma (FaR-RMS) Clinical Trial: A Report from the European Paediatric Soft Tissue Sarcoma Study Group (EpSSG)

**DOI:** 10.3390/cancers16050998

**Published:** 2024-02-29

**Authors:** Julia Chisholm, Henry Mandeville, Madeleine Adams, Veronique Minard-Collin, Timothy Rogers, Anna Kelsey, Janet Shipley, Rick R. van Rijn, Isabelle de Vries, Roelof van Ewijk, Bart de Keizer, Susanne A. Gatz, Michela Casanova, Lisa Lyngsie Hjalgrim, Charlotte Firth, Keith Wheatley, Pamela Kearns, Wenyu Liu, Amanda Kirkham, Helen Rees, Gianni Bisogno, Ajla Wasti, Sara Wakeling, Delphine Heenen, Deborah A. Tweddle, Johannes H. M. Merks, Meriel Jenney

**Affiliations:** 1Children and Young People’s Unit, Royal Marsden Hospital and Institute of Cancer Research, Sutton SM2 5PT, UK; henry.mandeville@rmh.nhs.uk; 2Children’s Hospital for Wales, Cardiff CF14 4XW, UK; madeleine.adams@wales.nhs.uk (M.A.); meriel.jenney@wales.nhs.uk (M.J.); 3Department of Paediatric Oncology, Gustave Roussy, 75015 Paris, France; veronique.minard@gustaveroussy.fr; 4Department of Paediatric Surgery, University Hospitals Bristol and Weston NHS Foundation Trust, Bristol BS1 3NU, UK; timothy.rogers@uhbw.nhs.uk; 5Department of Paediatric Histopathology, Royal Manchester Children’s Hospital, Manchester University NHS Foundation Trust, Manchester M13 9WL, UK; anna.kelsey@mft.nhs.uk; 6The Institute of Cancer Research, London SW7 3RP, UK; janet.shipley@icr.ac.uk (J.S.); ajla.wasti@rmh.nhs.uk (A.W.); 7Department of Radiology and Nuclear Medicine, University of Amsterdam, Amsterdam UMC, 1081 HV Amsterdam, The Netherlands; r.r.vanrijn@amsterdamumc.nl; 8Princess Máxima Center for Pediatric Oncology, 3584 CS Utrecht, The Netherlands; i.s.a.devries-10@prinsesmaximacentrum.nl (I.d.V.); r.vanewijk-2@prinsesmaximacentrum.nl (R.v.E.); b.dekeizer@prinsesmaximacentrum.nl (B.d.K.); j.h.m.merks@prinsesmaximacentrum.nl (J.H.M.M.); 9Birmingham Women’s and Children’s NHS Foundation Trust, Birmingham B15 2TG, UK; s.gatz@bham.ac.uk; 10Cancer Research UK Clinical Trials Unit, Institute of Cancer and Genomic Sciences, University of Birmingham, Birmingham B15 2TT, UK; c.m.firth@bham.ac.uk (C.F.); k.wheatley@bham.ac.uk (K.W.); p.r.kearns@bham.ac.uk (P.K.); wenyu.liu@ndph.ox.ac.uk (W.L.); a.j.kirkham@bham.ac.uk (A.K.); 11Fondazione IRCCS Istituto Nazionale Tumori, 20133 Milan, Italy; michela.casanova@istitutotumori.mi.it; 12University Hospital Copenhagen, DK-2200 Copenhagen, Denmark; lisa.lyngsie.hjalgrim@regionh.dk; 13Department of Paediatric Oncology, University Hospitals Bristol and Weston NHS Foundation Trust, Bristol BS1 3NU, UK; helen.rees@ubhw.nhs.uk; 14Department of Women and Children’s Health, University of Padova, 35122 Padua, Italy; gianni.bisogno@unipd.it; 15Alice’s Arc Charity, London EC1V 1AW, UK; sara.wakeling@alicesarc.org; 16KickCancer Foundation, 1000 Brussels, Belgium; delphine@kickcancer.org; 17Vivo Biobank, Translational & Clinical Research Institute, Newcastle University Centre for Cancer, Newcastle University, Newcastle upon Tyne NE1 7RU, UK; deborah.tweddle@newcastle.ac.uk

**Keywords:** rhabdomyosarcoma, clinical trial, chemotherapy, radiotherapy, randomisation, novel agents, FaR-RMS, EpSSG

## Abstract

**Simple Summary:**

This article summarises the international Frontline and Relapsed Rhabdomyosarcoma (FaR-RMS) clinical trial for patients with rhabdomyosarcoma. The trial has multiple research questions relating to chemotherapy and radiotherapy and biological and imaging studies as well as to the introduction of novel drugs for patients with very high-risk disease. The rationale, background, and international collaboration of the trial are explained, and how the data will be used to inform future studies is outlined.

**Abstract:**

The Frontline and Relapsed Rhabdomyosarcoma (FaR-RMS) clinical trial is an overarching, multinational study for children and adults with rhabdomyosarcoma (RMS). The trial, developed by the European Soft Tissue Sarcoma Study Group (EpSSG), incorporates multiple different research questions within a multistage design with a focus on (i) novel regimens for poor prognostic subgroups, (ii) optimal duration of maintenance chemotherapy, and (iii) optimal use of radiotherapy for local control and widespread metastatic disease. Additional sub-studies focusing on biological risk stratification, use of imaging modalities, including [^18^F]FDG PET-CT and diffusion-weighted MRI imaging (DWI) as prognostic markers, and impact of therapy on quality of life are described. This paper forms part of a Special Issue on rhabdomyosarcoma and outlines the study background, rationale for randomisations and sub-studies, design, and plans for utilisation and dissemination of results.

## 1. Introduction

The Frontline and Relapsed Rhabdomyosarcoma (FaR-RMS) clinical trial (ClinicalTrials.gov Identifier: NCT04625907) [1] is an overarching, multinational, study for children and adults with rhabdomyosarcoma (RMS). The trial, developed by the European paediatric Soft tissue sarcoma Study Group (EpSSG) and led by the Cancer Research UK Clinical Trials Unit at the University of Birmingham (UK), incorporates multiple research questions within a multistage design and an intention to utilise an adaptive approach for future amendments. The trial has three principal aims. These are to evaluate

the introduction of novel regimens compared to current standard of care in the most advanced disease states: Very High-Risk (VHR), High Risk (HR), and Relapse;the optimal duration of vinorelbine and cyclophosphamide maintenance chemotherapy;the use of radiotherapy to improve local control in VHR, HR, and Standard Risk (SR) patients and metastatic control in VHR disease.

In addition, the study evaluates

the risk stratification using *PAX-FOXO1* fusion gene status instead of histological subtyping;the use of [^18^F]FDG PET-CT and diffusion-weighted MRI imaging (DWI) response assessments as prognostic biomarkers for outcome following induction chemotherapy;the impact of local therapy (radiotherapy and surgery) on the health-related quality of life (HRQoL) for specific subgroups of patients.

The global trial commenced in 2020 and is currently open across UK, Europe, Australia, New Zealand, and Israel. This is the first multinational study for patients with RMS to include randomisations for patients of all ages with both newly diagnosed and relapsed disease and the first European trial to prospectively randomise patients to radiotherapy questions, and it incorporates a design which allows the introduction of emerging novel agents into frontline therapy throughout the timeframe of the trial. In addition, FaR-RMS incorporates sub-studies allowing the investigation of important factors, including molecular and radiological biomarkers, which may influence the design of future studies.

Patient and public involvement in the study, including representation on the Trial Management Group and Trial Steering Committee, and in developing a study logo [2], patient information materials, and patient-reported outcome measures (PROMS)/quality of life assessments has been integral to the study, and the study is actively supported by the parent-led charity Alice’s Arc [3].

## 2. Background

RMS is a rare soft tissue sarcoma (STS), with 59% of cases presenting in children and the rest occurring in adulthood, where the prognosis is poorer [4,5]. Although relatively rare, RMS is the most common paediatric STS, with a global incidence of 4–5 patients per million individuals aged <20 [5]. This equates to approximately 350 new cases a year in the US with a similar incidence reported in the UK [6] and Europe [7]. Pleomorphic RMS, which primarily occurs in older adults, is regarded as a different entity with a different clinical behaviour and therapeutic approach and is not included in this trial [8]. RMS can arise at many different sites throughout the body and is traditionally categorised into two major histological subgroups: alveolar (ARMS) and embryonal (ERMS) [9]. More recently, molecular characterisation of the presence or absence of a FOXO1 fusion is felt to be a more accurate method of risk stratification than histology and this theory will be tested in the FaR-RMS study [10]. In adults, 80% have histological diagnoses comparable to RMS in children, with a predominance of alveolar histology [5]. To date, very few clinical trials have included adult patients; however, a retrospective single-centre experience reported that treatment according to paediatric regimens may improve outcome [4].

RMS tumours are considered to be highly chemo-responsive; therefore, chemotherapy is an integral component of multi-modality therapy for RMS with a response rate (RR) of around 80–85% [11,12,13]. In newly diagnosed RMS, multi-agent chemotherapy regimens are assigned according to clinical risk factors [14,15]. The drugs used are combinations of long-established cytotoxic agents, including alkylating agents (ifosfamide and cyclophosphamide), vincristine, actinomycin D, and doxorubicin. Despite its relative chemo-sensitivity, local therapies such as radiotherapy and/or surgery are required to achieve optimal long-term local control. Incremental improvements in outcome have been achieved over the last three decades within clinical trials that have investigated stepwise modifications in the intensity and combinations of these drugs. In LR and SR disease, this has proven successful, with current 3-year EFS rates of 95% and 77%, respectively [15,16,17]. However, the greatest treatment challenges are for patients with HR and VHR (including metastatic) disease, as well as at the time of relapse, where improvement in survival with currently available agents has been inadequate; EFS remains below 70%, 45%, and 30%, respectively, and novel approaches are needed [14,18,19]. In addition, adults with RMS have outcomes inferior to children [8] and although teenage and young adult (TYA) patients benefit from being treated on paediatric protocols, they still have poorer survival outcomes than children, suggesting that a tailored treatment strategy may be warranted for these patients [20].

Several key clinical characteristics have been identified as having prognostic significance through previous EpSSG and other European RMS studies, including patient age, tumour size and site, histological subtype, nodal status, and stage. There has been an incremental improvement in survival outcomes for RMS over recent years [16], yet there remain several areas of unmet need, including patients with rare molecular changes such as MYOD1 alterations [21] and those with metastatic disease [22]. Patients with metastatic disease can achieve remission with intensive chemotherapy and local therapy in 75% of cases, but the vast majority still relapse, often at distant sites, resulting in a 3-year event-free survival (EFS) of only 27% [14,22]. At the time of relapse, RMS is generally refractory to treatment and has a 5-year overall survival (OS) of less than 20% [23].

In the last 20 years, a series of randomised clinical trials in paediatric RMS conducted by the three largest international collaborative paediatric oncology trial groups, namely the EpSSG and the Cooperative Weichteilsarkom Studiengruppe (CWS) group in Europe and the Children’s Oncology Group (COG) in North America, have defined current management strategies in RMS. EpSSG RMS 2005, the most recent EpSSG study, demonstrated EFS and overall survival OS rates of 70.7% and 80.4%, respectively, for patients up to 21 years with local or locoregional disease [15]. The COG ARST1431 intermediate risk RMS trial (similar to EpSSG HR patient cohort) is evaluating whether the addition of temsirolimus to the standard chemotherapy backbone (vincristine, actinomycin D, cyclophosphamide, and VAC) improves survival [24]. In the setting of relapsed disease, the recent European VIT-0910 randomised phase II trial has demonstrated the benefit of adding temozolomide to standard salvage backbone treatment of vincristine and irinotecan (VIrT) [25]. For patients with metastatic disease, the EpSSG MTS 2008 study demonstrated that EFS and OS rates are low for this group of patients, but that risk stratification can be undertaken using the Oberlin score to help identify subgroups with more favourable outcomes [22].

A key factor that has informed the design of the FaR-RMS trial has been early confidential sharing of data from the EpSSG studies with investigators from COG and CWS, facilitating combined analyses of rare subgroups, e.g., paratesticular [26] and parameningeal RMS [27], and more recently through the International Soft Tissue Sarcoma Consortium (INSTRuCT), which has developed a platform for international data sharing and collaborative analyses [28].

## 3. The Trial Design

The FaR-RMS study is an example of a Complex Innovative Design (CID) study [29] and is an “Umbrella” CID [30] that includes systemic therapy and local therapy research questions in multiple cohorts of patients with RMS, also incorporating research to increase our knowledge of the biology of RMS. The trial includes cohorts with newly diagnosed disease, stratified by clinical risk factors (see below), with relapsed disease, and, in patients with localised RMS, stratification for radiotherapy randomisations (RT1^B/C^) by risk of local failure. As a multi-arm, multistage (MAMS) platform study, its design incorporates both frontline and relapse components and uses an adaptive approach and a Bayesian statistical framework for parts of the study to allow efficient and seamless transition from one phase to the next, with the option to drop ineffective research interventions and/or introduce new ones during the lifetime of the study. This flexible approach within an overarching design allows efficiency of administration and cost of multiple relevant trial questions [29,30]. The overall trial schema is shown in Figure 1.

VHR = Very High-Risk; and HR = High Risk;I = Ifosfamide; V = vincristine; A = Actinomycin D; Do = Doxorubicin; I_R_ = Irinotecan; and R = Regorafenib;RT1^A^ = Randomisation for pre- or postoperative radiotherapy;RT1^B^ = Randomisation for dose escalation for patients with resectable tumours at high risk of local failure;RT1^C^ = Randomisation for dose escalation for patients with unresectable tumours at high risk of local failure;RT2 = Randomisation for radiotherapy to primary site vs. all metastatic sites for patients with widely metastatic disease (Oberlin score > 3);CT1^A^ = Randomisation between induction chemotherapy of IVADo and I_R_IVA in newly diagnosed patients with VHR disease;CT1^B^ = Randomisation between induction chemotherapy of IVA and I_R_IVA in newly diagnosed patients with HR disease;CT2^A^ = Randomisation between 12- and 24-month maintenance chemotherapy for patients with VHR disease;CT2^B^ = Randomisation between 6- and 12-month maintenance chemotherapy for patients with VHR disease;CT3 = Randomisation between VI_R_T and VI_R_R.

## 4. Eligibility and Risk Stratification

The FaR-RMS study is open to patients of all ages with a histologically confirmed diagnosis of RMS. Pleiomorphic RMS [31] is excluded. Patients may register on the study at different points during their frontline or relapse cancer pathway and must be registered on the study to be considered for participation in randomised questions; early registration is strongly encouraged. Within the study, clinical risk stratification is based on clinical features at first presentation including age, disease site, post-surgical clinical (IRS) group [32], tumour size, nodal status, and PAX: FOXO1 fusion status. This is modified from the previous EpSSG RMS 2005 risk stratification [15] as shown in Table 1. Key changes are as follows:PAX-FOXO1 fusion status is now used in place of alveolar/non-alveolar histology;The former subgroup D has moved from Standard to High-Risk;Genitourinary (GU) bladder/prostate and biliary sites are now considered favourable rather than unfavourable sites, based on analysis of RMS 2005 data;The newly designated VHR group now includes metastatic RMS in addition to PAX:FOXO1 fusion-positive, node-positive RMS (this differs from the previous EpSSG RMS 2005 study because metastatic patients are eligible for inclusion within FaR-RMS).

The Children’s Oncology Group includes *TP53* and *MYOD1* mutation status in its risk group assignment [33], and the EpSSG is currently considering whether these patients with adverse biology should be assigned to risk groups in a similar manner to those with fusion-positive RMS [21].

Patients in the LR and SR subgroups (A–C) can be offered trial registration and will be treated according to standard of care chemotherapy protocols although they are not eligible for randomised systemic therapy questions. Patients in subgroups C–H are potentially eligible for the randomised radiotherapy questions. All newly diagnosed patients will have confirmation of the *PAX-FOXO1* fusion status, central pathology review, and samples taken for future biological analysis and are eligible for sequential liquid biopsy studies of plasma molecular biomarkers. For patients with relapsed RMS, tumour biopsy at relapse is strongly recommended and they are eligible for the randomised relapse study, which is currently an investigator-led collaboration with Bayer and incorporates comprehensive biological and liquid biopsy assessments (see below).

In addition, FaR-RMS incorporates imaging sub-studies including an [^18^F]FDG PET study in frontline patients with baseline [^18^F]FDG PET-CT and DW-MRI studies. There are also patient-reported outcomes (PROMS) capturing health-related quality of life (HRQoL) assessments for the preoperative radiotherapy, metastatic radiotherapy, and relapsed RMS randomisations. Further PROMS/HRQoL research is being developed for other aspects of FaR-RMS.

The FaR-RMS trial is a comprehensive clinical research programme addressing the following objectives, which are summarised in Table 2:

### 4.1. Can Outcomes Be Improved by Utilising New Combinations of Systemic Anti-Cancer Therapies, including the Addition of New Biologically Targeted Drugs

#### 4.1.1. Frontline Treatment for Newly Diagnosed Patients 

For patients with HR, localised RMS, no induction chemotherapy combination has yet proven superior in efficacy to ifosfamide, vincristine, and actinomycin D (IVA) in Europe [12,19] or vincristine and actinomycin D and cyclophosphamide (VAC) in North America [16,17,34]. Although the toxicity profiles differ, no difference in outcomes was observed between VAC and IVA [16]. The current standard chemotherapy regimen for HR patients within EpSSG across Europe is IVA [19]. The North American ARST0531 study showed that VAC/vincristine-irinotecan had less hematologic toxicity and a lower cumulative cyclophosphamide dose with a similar outcome to patients receiving VAC. Further studies are needed to assess whether VAC/VI can be an appropriate chemotherapy backbone outside of the clinical trial setting [35].

For ARMS with involved regional lymph nodes (Group H in the EpSSG RMS 2005 risk stratification), which accounts for up to 10% of all RMS, an analysis of previous European co-operative studies suggested very poor survival (5-year EFS 39%), comparable to that of metastatic disease [36]. Based on improved outcomes reported in the SIOP MMT95 study (3-year EFS 57%) [12], in the EpSSG RMS 2005, Group H received intensified initial chemotherapy (IVADo: ifosfamide, vincristine, actinomycin D, and doxorubicin) and additional 6 months of maintenance chemotherapy with systematic local therapy to primary and nodal sites. With a median follow-up of 64.9 months (range 19.8–116.3), the 5-year EFS was 50% (95% CI 39%–59%) [37]. However, these studies included patients with fusion-negative ARMS, which are now understood to have a similar prognosis to ERMS (see below). In a recent analysis of patients treated within the RMS 2005 study, the 5-year EFS in fusion-positive, node-positive patients was 43% (95% CI 30–56%), compared with 74% (95% CI 54–87%) in fusion-negative (*p* = 0.01) patients, showing the need for improved treatments for this cohort of patients [38].

Metastatic RMS has a dismal prognosis with a 3-year EFS of 27% and an OS of 34% [22]. Treatment regimens have comprised combinations of IVA or VAC, with other agents showing evidence of activity in RMS (for example anthracyclines), given in a window setting. A good response early in treatment has, however, not resulted in a subsequent survival benefit. One study has demonstrated improvement in survival for a subset of metastatic patients with chemotherapy intensification [39]. The current EpSSG recommendation for induction chemotherapy for metastatic RMS is IVADo × 4 courses followed by IVA × 5 courses, based on the observed activity of single-agent doxorubicin in metastatic RMS [40]. A previous pharmaceutical company-sponsored study carried out by the EpSSG in collaboration with ITCC (Innovative Therapies for Children with Cancer), investigated the addition of the VEGF-targeted antibody bevacizumab to standard IVADo/IVA in patients with newly diagnosed, metastatic soft tissue sarcoma in children (BERNIE study). Unfortunately, bevacizumab did not show a significant improvement in EFS in either the whole group or the RMS subgroup [41]. However, the BERNIE study and the concurrent EpSSG MTS 2008 study showed improvement of outcome compared with historic controls overall; a pooled analysis of results showed that the 3-year EFS and OS were 35.5% (95% CI, 30.4 to 40.6) and 49.3% (95% CI, 43.9 to 54.5), respectively [22]. Whether this improvement was due to the intensive induction regimen including doxorubicin and/or the addition of maintenance treatment remains unclear due to the design of both studies.

Despite the lack of robust evidence of benefit for the addition of doxorubicin in the EpSSG RMS 2005 study for patients with localised disease, this approach has not been formally investigated in patients with alveolar, node-positive, or metastatic disease. Given the very poor survival for these patients, there is a reluctance to reduce therapy further (with the omission of doxorubicin), and within FaR-RMS IVADo remains the comparator arm for these patients. In view of the similarly poor outcomes for fusion-positive/node-positive RMS and metastatic RMS, these groups will be combined in the FaR-RMS trial to give a newly defined VHR group.

To try to improve systemic therapy in the frontline setting for patients in the HR and VHR groups (as defined above) and at relapse, the FaR-RMS trial will investigate the safety and efficacy of new systemic therapy combinations. The first new combination to be investigated builds on the promising activity of vincristine and irinotecan (VI_R)_ seen in window studies in metastatic RMS [42] and in relapsed RMS [43]. The ARST0531 trial compared VAC/VI_R_ with VAC with similar efficacy and less toxicity, resulting in the inclusion of irinotecan in frontline intermediate risk RMS in the adoption of VAC/VI_R_, as standard of care in intermediate and high-risk RMS trials by the COG group [35]. VI_R_ was also the standard arm in the EpSSG VIT-0910 study for patients with relapsed RMS [25]. A small study in 23 patients has demonstrated the feasibility of combining VI_R_ with IVA [44]. FaR-RMS includes a dose-finding phase Ib study combining a 5-day schedule of irinotecan with IVA (I_R_IVA) on days 8–12 of a standard 21-day cycle, and the recommended phase II dose of irinotecan will then be utilised in newly diagnosed HR and VHR patients in an upfront randomisation between the new I_R_IVA combination and either IVA (for HR patients) or IVADo (for VHR patients).

The study design allows new therapeutic agents to be introduced for evaluation, based on sound mechanistic biological and/or empirical evidence and prior clinical evidence from other settings and evidence of safety in phase 1 trials, as well as availability. Eligible patients will be entered into a Phase Ib component in limited ITCC centres with recognised expertise in undertaking paediatric oncology early-phase studies. The Phase Ib trial design is based on the Skolnik rolling 6 design [45], enrolling newly diagnosed patients with VHR disease to define the maximum tolerated dose of irinotecan in combination with IVA, collecting safety and preliminary activity data for the new agent combination.

#### 4.1.2. Patients with Relapsed Disease

In the most recent EpSSG RMS 2005 study, 29% of 1733 patients with localised disease had an event within 5 years [15] and 66% of 372 patients with metastatic RMS had an event within 3 years [22]. The outcomes following relapse of RMS are poor with fewer than 20% of patients salvaged overall [46], and better therapies are urgently needed. Prognostic factors at relapse are presenting characteristics, prior treatment, time to relapse, and pattern of relapse [18]. Although there is no internationally agreed standard approach to relapsed RMS, there is general agreement that biopsy confirmation of relapse, assessment of post-relapse prognosis, feasibility of further local control measures, and discussion of patient-centred goals are important for each patient [47].

The current European strategy for treatment of relapsed RMS is based on data from the COG group comparing two schedules of vincristine and irinotecan (VI_R_) in patients with a first relapse of RMS in a randomised phase II trial [43]. Within the EpSSG network, irinotecan in the 5-day VI_R_ schedule of vincristine, 1.5 mg/m^2^ on days 1 and 8 with irinotecan 50 mg/m^2^ days 1–5, formed the basis of a randomised phase II trial, VIT-0910, in relapsed and refractory RMS [25]. This trial evaluated the benefit of adding temozolomide to the standard salvage treatment of VI with a primary end point of objective response (OR). One hundred and twenty patients were enrolled (60 in each arm). The VI_R_T arm achieved significantly better Progression-Free Survival (PFS) (adjusted Hazard Ratio (HR) = 0.65, 95% Cl, 0.43–0.97) and OS (HR = 0.53, 95% CI, 0.33–0.83) compared to VI. VI_R_T is now considered standard therapy in Europe for patients at relapse who have already received alkylating agents in first line treatment and forms the control arm in the relapse randomisations within the FaR-RMS study. 

Regorafenib is a potent, oral multi-kinase inhibitor whose targets include vascular endothelial growth factor receptors (VEFGRs) 1, 2, and 3; tyrosine kinase with immunoglobulin and epidermal growth factor homology domain 2 (TIE2), platelet-derived growth factor receptors (PDGFRs), fibroblast growth factor receptors (FGFRs), c-KIT, RET, RAF-1, and BRAF (wild-type and V600E mutant). It is approved for the treatment of adult patients with metastatic colorectal cancer, gastrointestinal stromal tumours (GISTs), and hepatocellular carcinoma. The antitumor activity of regorafenib is thought to be mediated primarily by its antiangiogenic properties and accompanied by proapoptotic activity. In vitro studies have shown regorafenib causes a moderate growth inhibition of RMS cell lines and a significant tumour growth delay in vivo in all tumour models [48,49]. In preclinical models, complete regression was observed when regorafenib was combined with DNA-damaging agents such as irinotecan or radiotherapy in PDGFRA gene-amplified tumours but not in non-amplified ones [48,49]. FGFR1 and FGFR4may also be important targets in RMS [50,51].

The paediatric phase I study 15906 (REGOPEDS) of regorafenib in paediatric subjects with solid malignant tumours that were recurrent or refractory to standard therapy [52] demonstrated one transient partial remission and one disease stabilisation among the three RMS patients enrolled. The recommended phase 2 dose (RP2D) was defined as 82 mg/m^2^ once daily in a 3-weeks-on/1-week-off schedule. Toxicity was consistent with the adverse event (AE) profile seen in adults, apart from a higher incidence of grades 3/4 haematological toxicities in heavily pre-treated patients. Regorafenib exposure in children was in a similar range to that observed in adults, and a high between-subject variability was observed, with no apparent correlation with exposure by age.

To explore regorafenib in combination with chemotherapy, the REGOPEDS study was amended to test an escalating dose of regorafenib in combination with VI_R_ chemotherapy [53]. Two different dosing schedules were tested: vincristine 1.5 mg/m^2^ (days 1 and 8) and irinotecan 50 mg/m^2^ (days 1–5) were combined with daily oral regorafenib either on days 1–14 (concomitant schedule) or on days 8–21 (sequential schedule) in a 21-day cycle. Concomitant dosing was discontinued when several grade 3 dose-limiting toxicities were reported in the first two patients treated on this schedule (peripheral neuropathy and liver injury, pain, vomiting, and febrile aplasia). The maximum tolerated dose and recommended phase 2 dose of regorafenib in the sequential schedule was 82 mg/m^2^. Radiological responses were observed in 7 of 12 patients with both alveolar and embryonal RMS (1CR and 6PR), including patients who previously received irinotecan chemotherapy. Two patients remained on treatment for more than 1 year. Overall, the VI_R_R regimen has shown reassuring preliminary activity in a relapsed/refractory RMS patient population. Safety and toxicity signals in the VI_R_R combination indicate that the toxicity level is in the range of the VI_R_T combination. The level of activity seen for the VI_R_R combination was considered sufficient and worth proceeding to the randomized phase II stage against standard VI_R_T chemotherapy, and this is being taken forward within the FaR-RMS in partnership with Bayer, the manufacturer of regorafenib.

Regorafenib has both tablet and granulate formulations with comparable bioavailability; however, these are currently not widely available and are not recommended for use outside of a clinical trial. In the FaR-RMS CT3 randomisation, regorafenib will be combined with the VI_R_ schedule tested in REGOPEDs at a dose of 120 mg for patients aged 12 years and over and a weight of ≥40 kg; 82 mg/m^2^ for patients > 2 years and <12 years weighing < 40 kg; and 65 mg/m^2^ for patients aged 6 months–2 years in a seamless phase II/III study design.

The FaR-RMS CT3 randomisation is an example of how a CID study such as FaR-RMS provides opportunities for partnership with the pharmaceutical industry to address important clinical questions. FaR-RMS was proposed to Bayer as a platform for the introduction of VI_R_R in the relapse setting. FaR-RMS will provide high-quality data to support Bayer’s European Medicines Agency-approved Paediatric Investigation Plan (PIP). For Bayer, access to a sufficient number of relapsed patients with this rare cancer is only possible in the context of an international, academic–industry collaborative clinical trial.

### 4.2. Improving Outcomes for RMS through Optimisation of Radiotherapy Strategy

#### 4.2.1. The Timing of Adjuvant Radiotherapy

Historically, radiotherapy for RMS has been delivered after surgical resection. However, preoperative radiotherapy has a number of potential advantages over postoperative radiotherapy: The accuracy in defining the radiotherapy field is improved because the intact tumour target volume is easier to define; the residual tumour may act as a form of ‘spacer’, meaning that less uninvolved normal tissue is exposed to the higher radiotherapy dose; a significant proportion of the irradiated tissue will be removed surgically, which may reduce the risk of second tumours; there is a biological rationale as the tumour and surrounding tissues are less hypoxic than in the postoperative setting, and hypoxia increases tumour radio-resistance [54]. In other soft tissue sarcomas (STSs), preoperative radiotherapy has been increasingly used in standard clinical settings. O’Sullivan [55] showed a small significant improvement in OS in adult patients with extremity STS randomised to receive preoperative radiotherapy at 50 Gy compared to postoperative radiotherapy at 66 Gy, although this was counterbalanced by an increased risk of acute wound complications. Preoperative radiotherapy is being investigated in a number of STS studies in the US, including NCT01344018 and NCT02180867. There is limited published experience on preoperative radiotherapy for RMS: a cohort of 17 patients with bladder–prostate RMS in the German CWS96 study had a reported a 5-year EFS of 82% [56]. In the FaR-RMS trial, the efficacy (local control), safety, and impact on health-related quality of life (HRQoL) of preoperative radiotherapy compared to standard postoperative radiotherapy will be investigated in the RT1^A^ randomisation, which is open to patients with predicted R0 or R1 resection who require radiotherapy in addition to surgical resection. Patients will be randomised in a 1:1 ratio irrespective of disease site. Patients who require brachytherapy will be excluded. The trial team understands that there may be local provider preferences for pre- or postoperative radiotherapy but would recommend randomisation due to the lack of definitive evidence in RMS patients.

#### 4.2.2. Can Dose Escalation of Radiotherapy Improve Local Control in Patients at a Higher Risk of Local Failure?

The current strategy for radiotherapy has been established over the last 40 years in European and US collaborative group studies. Doses ranging from 36 to 55 Gy (conventionally fractionated) and 59.4 Gy (hyperfractionated radiotherapy: HFRT) have been employed. In the SIOP MMT studies, 45 Gy was the recommended dose, plus 5 Gy for microscopic residual or 10 Gy for macroscopic residual disease [57]. The true impact of dose escalation for RMS patients where there is a higher local failure risk has not been adequately investigated. To date, only the COG IRS IV study has asked a randomised radiotherapy question comparing HFRT (59.4 Gy in 54 × 1.1 Gy twice-daily fractions) with 50.4 Gy conventional fractionation (1.8 Gy once daily). This study showed no difference in the local control suggesting that the biological effective dose (BED) for tumour control with 59 Gy, when delivered in this hyperfractionated schedule (using a low dose per fraction), was similar to 50.4 Gy delivered using conventional fractionation, and, in fact, there had not been a true radiotherapy dose escalation [58]. Increased acute toxicities were observed in the HFRT arm, and therefore conventional fractionation remains the gold standard for RMS.

The potential benefits of radiotherapy dose escalation in RMS still need to be determined. In the IRS II–IV studies [59], patients with macroscopic disease after the first surgery received <47.5 Gy radiotherapy; a higher rate of local failure of 35% was observed for tumours ≥ 5 cm size compared to 18% for tumours < 5 cm. Yet patients who received >47.5 Gy had a lower local failure rate of 15%, irrespective of tumour size. Size ≥ 5 cm was also identified as a key factor increasing the risk of local failure in the COG D9803 study [60]. In an unpublished multivariate analysis from the RMS 2005 study, IRS Group 3 patients with localised disease up to the age of 21 years demonstrated only unfavourable sites to be associated with a higher local failure risk (HLFR); in this analysis size > 5 cm was not an independent risk factor for local failure. As both the acute and late toxicities of radiotherapy are known to increase when higher doses of radiotherapy are used, it is important to identify those at a higher HLFR where the benefits of improved tumour control potentially resulting from radiotherapy dose escalation are more likely to outweigh the potential consequences.

Adult patients are known to have worse outcomes, including local failure, but to date have been excluded from the majority of collaborative group RMS studies. Patients with resectable tumours with an HLFR (defined as unfavourable disease site (see Table 1) or adult patients), will be eligible for randomisation to receive either standard dose radiotherapy 41.4 Gy versus dose-escalated radiotherapy 50.4 Gy (RT1^B^), with the additional 9 Gy for dose-escalated patients delivered to the extent of tumour remaining after three cycles of induction chemotherapy. Patients with unresectable disease with a complete response following induction therapy will not be eligible to enter a radiotherapy trial question. Patients whose tumour is not suitable for surgical resection, with an incomplete response following induction therapy, and where there is an HLFR, may be randomised between standard dose radiotherapy 50.4 Gy versus dose-escalated radiotherapy 59.4 Gy (RT1^c^). Figure 2 outlines the radiotherapy randomisations for the local disease site.

#### 4.2.3. Can Radiotherapy to All Metastatic Sites in Unfavourable Metastatic Disease Reduce the Risk of Relapse and Improve EFS?

There are conflicting data as to whether radiotherapy to metastatic sites truly influences outcomes for RMS. To date, the standard of care for metastatic RMS has been systematic irradiation of all metastatic sites whenever feasible (MTS-2008 registry study for metastatic RMS within RMS 2005) [61], in sharp contrast to guidelines for adult STS where radiotherapy to metastatic sites has not been the standard of care. In the COG studies, patients with >3 metastatic sites are categorised as having extensive metastatic disease and radiotherapy is delivered at week 20. Radiotherapy for these patients is challenging, and COG advises that certain metastatic sites are prioritised, leaving other sites where radiotherapy may need to be omitted or delivered later at week 47. However, these guidelines have been open to interpretation and the randomised BERNIE study, evaluating bevacizumab in combination with standard chemotherapy, showed that of 102 metastatic RMS patients, only 31 had radiotherapy to all sites, 49 had radiotherapy to some sites (partial radiotherapy), and 22 had no radiotherapy; OS was improved in those receiving radiotherapy although selection bias could have contributed to this [41]. A small single-centre series of 13 patients with metastatic RMS or Ewing sarcoma (EWS) receiving systematic radiotherapy (>40 Gy) to all metastatic sites reported a local control rate for metastases of 92% and OS of 35%, both at 5 years [62]. A further series of six patients with metastatic RMS, treating all metastases with radiotherapy (41.4 Gy–50.4 Gy), achieved 100% local control, yet out-of-field relapses were seen in 50%, and median OS was only 31.8 months [63].

For patients with lung-only metastases (approximately 28% of patients with metastatic disease), the evidence of a benefit for whole-lung radiotherapy is also mixed. A retrospective analysis of 46 patients from the IRS IV study reported that 25 received whole-lung radiotherapy and 16 did not, with the treatment strategy determined by the treating centre with no randomisation; those receiving lung radiotherapy had fewer lung recurrences, but the difference in OS (47% vs. 31%) was not significant [64]. A report from CWS on 29 patients with ERMS and lung-only metastases showed a complete response to induction chemotherapy in 22 [65]. Ten patients received local therapy (nine whole-lung radiotherapy and three metastatectomy); however, nineteen patients did not, without any apparent effect on OS, EFS, or the rate of local relapse in the lungs. In a recent analysis of 59 patients with RMS and lung-only metastases, from the EpSSG MTS 2008 protocol, those receiving lung radiotherapy had a superior 3-year EFS (RT, n = 26, EFS 56%, 95% CI 35–73%; no RT, n = 24, EFS 33%, 95% CI 16–52%, *p* = 0.0435) [66].

Apart from the lack of clear evidence that radiotherapy to all sites including metastases is effective, it can have an adverse impact on HRQoL in a patient group with a dismal prognosis and can produce myelosuppression, limiting the delivery of further chemotherapy. The multivariate pooled analysis from US and European cooperative groups, published in 2008, has defined the following prognostic (Oberlin) factors for RMS patients with metastatic disease [14]:○Age < 1 y or ≥10 y;○‘Unfavourable’ site: extremity, other, and unidentified;○Bone or bone marrow involvement;○≥3 metastatic sites.

In this analysis, there was a clear separation in EFS between groups: patients with ≤1 risk factor having a favourable 3-year EFS of 44% whereas those with ≥2 prognostic factors having a more unfavourable outcome with a 3-year EFS of only 14%. FaR-RMS aims to investigate whether radiotherapy to metastatic sites improves survival for patients with unfavourable metastatic RMS and to evaluate the effects on HRQoL of this treatment.

### 4.3. Can Prolongation of Maintenance Therapy Reduce the Risk of Relapse and Improve OS?

The rationale for utilising maintenance chemotherapy in RMS and results from both EpSSG and other available studies are reviewed in detail in a separate paper in this Cancers Special Issue [67].

RMS usually responds well to initial chemotherapy, and CR or nearly complete PR can be achieved with multimodality therapy. The challenge is thus to maintain disease remission by eliminating minimal residual disease. Approaches with longer low-dose treatments, so-called maintenance or metronomic chemotherapy, have been developed. In addition to proven anti-angiogenic activity, other potential mechanisms of action have been proposed, such as restoration of anti-cancer immune response and induction of tumour dormancy [68,69]. Two phase II studies in relapsed/refractory RMS patients combined intravenous vinorelbine with continuous daily oral cyclophosphamide (VnC), resulting in promising response rate rates of 36 and 37% [70,71].

In the EpSSG RMS 2005 trial, patients with HR disease in clinical complete remission at the end of standard induction treatment were randomised to stop therapy or to receive a 6-month prolongation of treatment with maintenance therapy comprising intravenous vinorelbine 25 mg/m^2^ on days 1, 8, and 15 of each 28-day cycle with continuous daily oral cyclophosphamide 25 mg/m^2^ (VnC). In the intention-to-treat population, 5-year disease-free survival was 77.6% (95% CI: 70.6–83.2%) with maintenance chemotherapy versus 69.8% (95% CI: 62.2–76.2%) without maintenance chemotherapy (Hazard Ratio [HR] 0.68 [95% CI 0.45–1.02]; *p* = 0.061), and 5-year overall survival was significantly improved at 86.5% (95% CI: 80.2–90.9%) with maintenance chemotherapy versus 73.7% (95% CI: 65.8–80.1%) without (HR 0.52 [95% CI 0.32–0.86]; *p* = 0.0097). The toxicity of maintenance therapy was manageable, with mainly hematologic toxicity and infections (31% grade 3, no grade 4). Importantly, the median time to relapse calculated from the randomisation date to the event was 6.9 months (IQR 3.0–16.1) in the group given no further treatment and 10.1 months (IQR 6.9–15.4) in the maintenance chemotherapy group. Because most events in the maintenance chemotherapy group occurred after 6 months of VnC maintenance had been completed, this supports the new randomisation between continuing maintenance treatment for a further 6 months (12 months total) versus stopping treatment after 6 months of standard maintenance chemotherapy within FaR-RMS. In FaR-RMS, the optimal duration of VnC maintenance therapy in VHR RMS will be evaluated in a randomisation to stop treatment after 12 months or to continue for a further 12 months (total 24 months).

Oral vinorelbine is widely used in adults with acceptable and reliable pharmacokinetic profiles at clinically relevant dosage levels. In adults, oral vinorelbine has approximately 40% bioavailability; thus, a dose of 60 mg/m^2^ orally is the equivalent of 25 mg/m^2^ i.v. [72]. In a previous Phase II study of vinorelbine and continuous low-dose cyclophosphamide in children and adolescents with a relapsed or refractory malignant solid tumour, bioequivalence data demonstrated that both Body Surface Area (BSA)-standardized clearance and total drug exposure following 25 mg/m^2^ i.v. vinorelbine were equivalent between children >4 years and the adult series [71]. Conflicting results have been reported by COG in a Phase I study in paediatric cancer patients with oral (week 1) and i.v. (weeks 2 to 6) vinorelbine) [73]. Higher mean intravenous total body clearance was observed compared with adult reports, and mean oral bioavailability was 28.5 ± 22.5% with the apparent oral clearance and volume of distribution higher than in adults given similar oral doses.

Despite conflicting results regarding a PK analysis in children/adolescents, full oral maintenance including oral low-dose cyclophosphamide and oral vinorelbine will be an option for patients with VHR disease in the FaR-RMS study. Oral vinorelbine will provide patient convenience and better patient acceptability in the context of prolonged VnC maintenance. For young patients (<4 years) and patients with difficulty swallowing tablets or capsules, the intravenous vinorelbine formulation can be considered and further age-adapted oral vinorelbine formulations should be developed. Additionally, further vinorelbine (and cyclophosphamide) PK studies as well as monitoring of systemic and immune effects of maintenance therapy will be conducted in FaR-RMS. Overall, FaR-RMS will help to establish the optimal duration of VnC maintenance chemotherapy for both HR and VHR RMS.

### 4.4. To Assess Whether PAX-FOXO1 Fusion Status in place of Histological Diagnosis Improves Riskt Stratification

There are two main histological subtypes of RMS, ARMS accounting for ~30% of RMS and ERMS [47]. ARMS is characterised by the presence of a PAX 3: or less commonly PAX 7:FOXO1 gene fusion, which is present in around 80% of ARMS tumours. PAX3/7:FOXO1 fusion gene-negative ARMS is clinically and biologically similar to ERMS, and PAX3:FOXO1 fusion is a key prognostic indicator in RMS [74]. The molecular distinction between PAX3/7:FOXO1 fusion gene-positive and -negative groups has been modelled as superior for risk stratification to the histological subtype [10,75]. Based on work from several groups [76,77], this is now incorporated into the risk stratification in the FaR-RMS trial and the same approach is also now used by the Children’s Oncology Group. The approach and methodology to determine PAX3/7:FOXO1 status have been reviewed in detail elsewhere [78]. Risk stratification using PAX3/7:FOXO1 status is expected to reassign approximately 7% of patients, mostly to reduce treatment intensity (in fusion-negative ARMS), with the potential associated benefit of reducing toxicities in these patients [77]. The impact of using fusion status in risk stratification is being assessed prospectively in FaR-RMS, where it is estimated to be feasible to evaluate based on the expected numbers of patients [76].

## 5. Statistical Considerations

Trial recruitment is planned to last for 7 years. The primary endpoint for the phase Ib study is a recommended phase II dose, and for chemotherapy and RT2 randomisations, it is EFS. The primary endpoint for RT 1a, b, and c is local failure-free survival. All randomisations will be analysed using a Bayesian approach according to a detailed statistical analysis plan. The sample size is pragmatic and based on the number of patients that can be recruited in Europe over the trial’s accrual period. Given the difficulties of performing realistic sample size calculations with small patient numbers, each randomisation will be treated as independent. Planned sample sizes are shown in Table 3.

## 6. Health-Related Quality of Life and Patient-Reported Outcome Measures

Patients with RMS, in particular those with HR disease, undergo highly intensive therapies in the attempt to cure their disease. This, along with the physical and emotional morbidity associated with the cancer itself is likely to adversely affect the health-related quality of life (HRQoL) of both the patients and their families.

As well as measuring conventional outcomes such as EFS and OS, increasing attention is now given to patient-reported outcomes (PROs), defined as ‘any report of the status of a patient’s health condition that comes directly from the patient, without interpretation of the patient’s response by a clinician or anyone else,’ in order to evaluate treatment efficacy [79]. PROs include a range of outcomes such as symptoms, physical functioning, and HRQoL [80].

To date, there are only a limited number of published studies of HRQoL during treatment for RMS [81,82]. These have focused on particular disease sites or specific treatment modalities such as brachytherapy or proton beam radiotherapy. The large cohort within the FaR-RMS trial presents an opportunity to study HRQoL across a wide range of disease sites and compare HRQoL scores between different treatment strategies.

A key secondary aim for the FaR-RMS randomisations evaluating the impact of radiotherapy timing, and also for metastatic radiotherapy, is to study and better understand HRQoL of patients and identify whether there are any differences in HRQoL between the different radiotherapy arms of the study. If no difference in survival outcomes, or toxicities, are seen between different randomisation arms, then HRQoL outcomes may be key in determining which treatment strategies to take forward as standard of care.

All patients taking part in the preoperative versus postoperative radiotherapy randomisation (RT1A) and radiotherapy to all disease sites versus loco-regional radiotherapy randomisation in patients with unfavourable metastatic disease (RT2), will be provided with the appropriate HRQoL questionnaires (where the appropriate language questionnaire is available). This comprises the PedsQL generic and cancer-specific version [83] for children aged <18 years and the EORTC QLQ-C30 [84] for patients aged >18 years. In addition, limited PROs about tolerability of different regorafenib formulations will be collected in the current relapse randomisation. The full rationale and details of the PROMs study is outlined in a separate paper, which forms part of this RMS Special Issue series [85].

## 7. Imaging Studies

Radiological imaging is key in the diagnosis, definition of local extension, disease staging, response assessment, and follow-up of RMS. In the FaR-RMS study, MRI is the preferred modality of imaging of the primary tumour and [^18^F]FDG PET-CT is the recommended modality for examining bony involvement, nodal involvement, and distant metastases. Early anatomical tumour size or volume response (one-, two-, or three-dimensional) after neo-adjuvant chemotherapy is not a valid early surrogate marker of long-term survival [8], and to evaluate the efficacy of new treatments, we are using EFS and OS as endpoints. As a result, trials like FaR-RMS now take 7–10 years to achieve adequate patient accrual and follow-up. In the FaR-RMS trial, prospective imaging study questions are included with the aim of validating potential new early surrogate markers of outcome in RMS. [^18^F]FDG PET-CT and diffusion-weighted MRI (DW-MRI) as functional markers of response have shown to be surrogate markers in solid tumours [86,87], but only limited data are available from retrospective series in RMS [86,88,89,90].

In the FaR-RMS trial, two imaging questions regarding the value of early tumour response after three cycles of neo-adjuvant chemotherapy are being evaluated: (i) Is response measured according to the PET Response Criteria for Solid Tumours [91] by [^18^F]FDG PET-CT an early prognostic marker of survival? (ii) Is change in the apparent diffusion coefficient (ADC) evaluated by DW-MRI an early prognostic marker of survival?

The primary classification of FDG-PET response is with the use of PERCIST [92,93], including quantitative PET indices (standard uptake values (SUVs)). Secondary objectives include visual assessment with the Deauville score [94]. The primary variable of the DW-MRI sub-study is the absolute change in the mean ADC, as compared to baseline assessment. Secondary variables include multiple DW-MRI indices, including an ADC histogram analysis. The FDG-PET study is available to all patients who have had PET-CT at baseline except those stratified as low risk according to the FaR-RMS risk stratification. The DW-MRI study includes patients with both localized or metastatic disease who have gross residual disease after an incomplete primary resection or biopsy.

Unique to this international multicentre study is that all imaging data will be centrally collected, providing the possibility for imaging quality assessment, standardized analysis, expert review, and in combination with clinical data, the application of radiomics [95,96]. The image repository was developed as part of the European Society of Paediatric Oncology (SIOPE) Quality and Excellence in Radiotherapy and Imaging for Children and Adolescents with Cancer across Europe in Clinical Trials initiative (QUARTET) [97]. To improve the consistency of data collection in this rare tumour, the EpSSG, European Society of Paediatric Radiology and CWS published an imaging guideline for RMS [98]. Additionally, for the FDG-PET study, sites are strongly recommended to adhere to available international guidelines [99] and for the DW-MRI study, quality site visits are planned by the study team to optimise DW-MRI data acquisition.

## 8. Associated Biological/Biomarker Studies + Opportunity for Biobanking

### 8.1. Frontline

Recent insights into the genetics and molecular biology of RMS have provided data critical to accurate diagnosis and risk stratification. However, further biological studies are required to increase the molecular understanding of disease processes that will lead to improvement in the unacceptable short- and longer-term outcomes for poor-risk RMS patients. Historically, it has been challenging to internationally coordinate efforts geared towards the systematic collection of biospecimens in RMS, given that no single infrastructure or sufficient funding has been available to facilitate this. Individual centres have collected biospecimens that have been used in research but largely in a siloed manner.

Because FaR-RMS is on target to recruit approximately 200 frontline patients per year across participating countries for the duration of the trial, it offers an unparalleled opportunity for the systematic collection of samples for research. Biological sample collection should be a key element of clinical trials, and FaR-RMS provides a unique opportunity as samples will be clinically annotated, and outcome-related data will continue to mature over time. Research using such samples is critically important for the robust validation and identification of correlations among novel biomarkers, molecular characteristics, and clinical parameters, which include response to treatment. Materials to be collected include tumour samples (including at relapse), plasma, and cell fractions from bone marrow and blood. This may be up to ~10 samples from each patient from diagnosis and through treatment at defined time points specific to the risk group.

Sample collection through FaR-RMS has the potential to improve risk stratification and identification of patients undergoing treatment who are at the highest risk of relapse. Earlier identification of poor-risk patient groups could enable more prompt therapeutic intervention, be that in the form of more intense conventional therapies or through the testing of novel, molecularly tailored approaches to treatment. Molecular analyses of samples and use of derived models is also integral to preclinical research that should underpin new treatment strategies. FaR-RMS therefore presents us with a comprehensive platform to enable robustly powered and coordinated current and future biological investigations that will improve outcomes for RMS patients.

### 8.2. Relapse (Collaboration with Bayer)

The collaboration with Bayer for the CT3 relapse question has also provided the opportunity for comprehensive biomarker analysis in relapsed patients, supported by Bayer. In addition to DW-MRI imaging analysis at baseline and reassessment time points, comprehensive molecular profiling of relapsed tumours (and from first diagnosis, where available) and plasma molecular biomarker analysis, including ctDNA analysis, will be undertaken for all patients entered into the relapse study.

### 8.3. VIVO Tissue Bank

In order to capitalise on the potential for important biological studies within FaR-RMS, the aim is to create a sample and data resource that can support RMS research. This has been established through a pathway, which considers logistics and sample prioritisation for research. The VIVO Biobank, which was preceded by the Children’s Cancer and Leukaemia Group (CCLG) Tissue Bank for solid tumours and the Blood Cancer UK Cell Bank for haematologic malignancies, is a new collaboration between Cancer Research UK and Blood Cancer UK. It is UK Health Research Authority (HRA)-approved and is the largest research resource in the UK which stores biological samples and data from children and young people affected by cancer. The platform facilitates meeting multi-stakeholder research objectives through a centralised model of biobanking.

Among the strengths of this centralised approach is inclusivity, with the ability to bank samples from centres that do not have independent institutional or national biobanking initiatives. This simultaneously increases the number of samples banked with availability of associated data that will future-proof subsequent research. Seamless, centralised management of samples including a nominal payment/sample, enables coordination, collaboration, and tissue prioritisation with efficiencies in the central management of samples and data generated under an overarching agreement. Together, this maximizes potential benefit from the trial for future patients and meets patient and parent expectations around sample collection for future research.

The FaR-RMS TMG has a formal agreement with the VIVO Biobank so that research samples collected from FaR-RMS patients are stored and managed by the VIVO Biobank. These samples are ring-fenced for use by FaR-RMS trial-linked research studies. A process has been agreed via the FaR-RMS TMG and the VIVO Biobank Sample and Data Access Committee for approval of biological studies that maximises potential value of available material and is open to the RMS clinical and research community.

For countries that are unable or do not wish to collect samples which will contributed to the VIVO Biobank, it is recommended that samples are collected and stored in accordance with national biobanking and molecular profiling initiatives for future biological studies.

## 9. Conclusions and Future Perspectives

In this article, we have described the background to the research questions that have been included within the FaR-RMS study, which is currently recruiting as a multinational EpSSG study in collaboration with the Cancer Research UK Clinical Trials Unit, Birmingham, UK. As described, this CID study provides an opportunity to simultaneously address multiple current areas of unmet need and knowledge gaps in the systemic and local treatment of RMS and will provide prospective tumour and liquid biopsy collection in both newly diagnosed and relapsed patients, with clinical annotation. The biobanking of samples will also create an important sample repository for future use to further our understanding of rhabdomyosarcoma biology, circulating plasma biomarkers, and tumour evolution.

The flexibility of the MAMs study design within FaR-RMS provides the opportunity to evaluate the benefits of new systemic therapy combinations efficiently. The design allows important collaborations with the pharmaceutical industry to accelerate access to new agents for patients with high-risk or metastatic disease with potential efficient future implementation in the standard treatment of RMS.

The EpSSG group continues to work with international colleagues to share data and best practice, discuss areas of uncertainty, and ensure that the FaR-RMS study complements and enhances knowledge gained in other settings. 

## Figures and Tables

**Figure 1 cancers-16-00998-f001:**
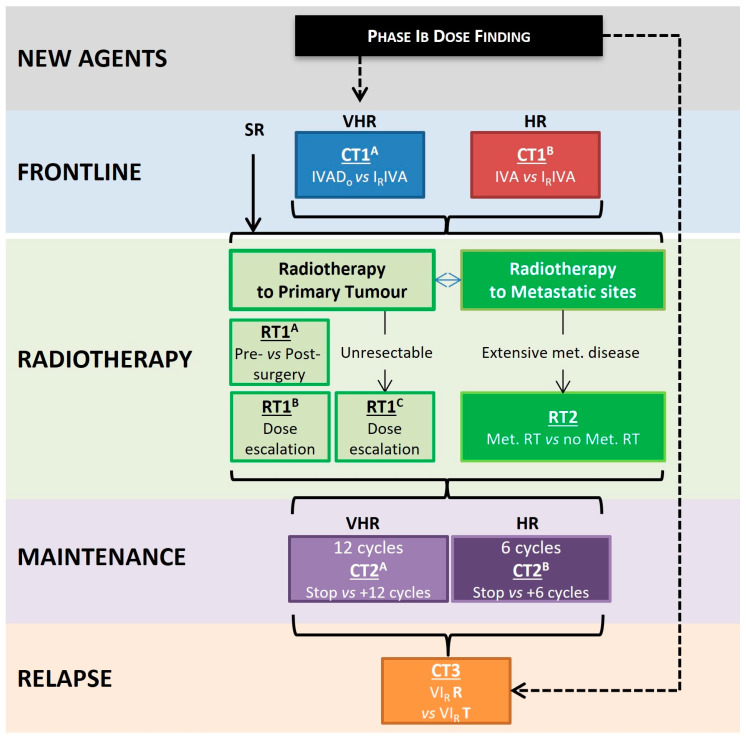
Trial schema for FaR-RMS clinical trial.

**Figure 2 cancers-16-00998-f002:**
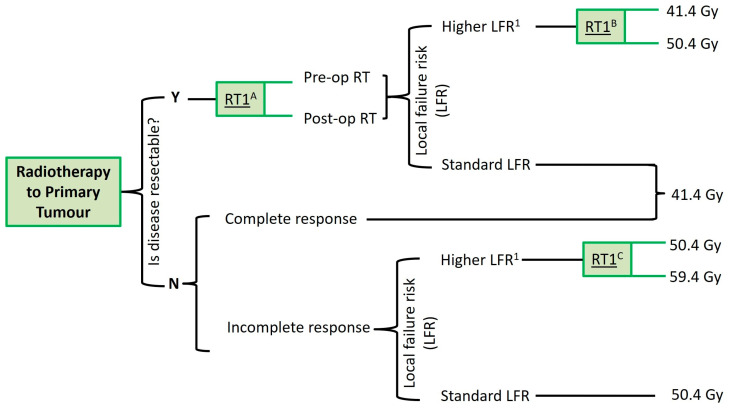
Schema for radiotherapy randomisations for local disease site. ^1^: Higher local failure rate (LFR) defined as unfavourable site of patient aged ≥18 years.

**Table 1 cancers-16-00998-t001:** Risk stratification as determined at diagnosis.

Risk Group	Subgroup	Fusion Status	IRS Group	Site	Nodal Status	Size or Age
Low-risk	A	Negative	I	Any	N0	Both favourable
Standard risk	B	Negative	I	Any	N0	One of both favourable
	C	Negative	II, III	Favourable	N0	Any
High-risk	D	Negative	II, III	Unfavourable	N0	Any
	E	Negative	II, III	Any	N1	Any
	F	Positive	I, II, III	Any	N0	Any
Very high-risk	G	Positive	II, III	Any	N1	Any
	H	Any	IV	Any	Any	Any

*Fusion status*: Where fusion gene status is unavailable, histopathology will be used. Non-alveolar disease should be defined as fusion gene-negative, and alveolar disease should be defined as fusion gene-positive. *Site*: Favourable sites are GU, including bladder–prostate, head and neck non-parameningeal, orbit and biliary primaries. Unfavourable sites are all other sites. *Node stage*: N0 = 0 positive lymph nodes and N1 = ≥ positive lymph node. *Age*: Favourable is defined as age over 1 and under 10 years of age at diagnosis. *Size*: Favourable primary tumour is ≤5 cm as the longest diameter, and patients that are assessed as not evaluable will be included in >5 cm group. IRS Group [32].

**Table 2 cancers-16-00998-t002:** The trial questions.

Question	Description
1	Can outcomes be improved by utilising new combinations of systemic anti-cancer therapies, including the addition of new biologically targeted drugs in: Frontline treatment for newly diagnosed patients? Patients with relapsed disease?
2	Can outcomes be improved through optimising radiotherapy schedules? Is there any benefit in delivering adjuvant radiotherapy preoperatively rather than postoperatively? Can dose escalation of radiotherapy improve local control in patients at a higher risk of local failure? Can radiotherapy to all metastatic sites in patients with unfavourable metastatic disease reduce the risk of relapse and improve EFS?
3	Can prolongation of maintenance therapy reduce the risk of relapse and improve OS for patients with HR and VHR disease?
4	Can PAX-FOXO1 fusion status be utilised instead of histological diagnosis to improve treatment stratification?
5	Can [^18^F]FDG PET-CT and the apparent diffusion coefficient (ADC) evaluated by the DW-MRI response assessment following induction chemotherapy be used as prognostic biomarkers for local control and/or survival?
6	Can the DWI-MRI response assessment following induction chemotherapy be used as a prognostic biomarker for local control and/or survival?

**Table 3 cancers-16-00998-t003:** Sample sizes for trial randomisation questions.

Randomisation		Minimum Number of Patients in Total	Assumed Baseline Event Free Rate for the Primary Outcome, 3-Year (%)
Radiotherapy	1a	350	80
	1b	315	79
	1c	350	72
	2	210	40
Newly diagnosed chemotherapy	Very high-risk	370	35
	High-risk	470	65
	Very high-risk maintenance	260	35 to 45
	High-risk maintenance	240	65
Relapse		260 for the regorafenib questions 420 in 7 years with additional arms	30, 1 year

## Data Availability

Not applicable.

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
