# Peer review of "Frontline and Relapsed Rhabdomyosarcoma (FAR-RMS) Clinical Trial: A Report from the European Paediatric Soft Tissue Sarcoma Study Group (EpSSG)"

_cancers, 2024, doi:10.3390/cancers16050998_

Round 1

Reviewer 1 Report

Comments and Suggestions for Authors

This is a very well written review of the FaR-RMS clinical trial. I enjoyed reading it.  It is very thorough and a comprehensive review of the science behind the study design.  . My only concern is that it is quite long and could benefit from abbreviation, focusing more on explaining the design of the study and a little less on the background and rationale in sections, especially in the latter half of the manuscript.  This is minor.

Line 337 - omit this paragraph?

Line 387 - omit this paragraph?

Line 396 - omit this paragraph?

Line 404: - abbreviate this section and explain in more depth the actual randomization (only the last sentence actually explains it). is it randomized? is it 1:1 randomization? Is there provider discretion? 

Line 427 - shorten this section and better explain the RT2B and RT2C randomizations of the study

Lines, 467, 518, 578, 596, 636, 712 - all section could b shortened. 

Author Response

Thank you

Reviewer 2 Report

Comments and Suggestions for Authors

This manuscript reports the design of the FAR-RMS study, as part of a special collection of rhabdomyosarcoma treatment approaches by the EpSSG. The study is a complex international study that attempts to answer multiple questions in the RMS field, ranging from risk stratification, response assessment, frontline chemotherapy approach, radiation strategy, maintenance strategy, and salvage therapy.  

Major comments:

There are numerous simple grammatical errors that appear to be carry-overs from previous edits, which make it rather difficult to read in some sections. If a non-English major like myself can pick those up so easily, there needs to be additional quality control here.

Figure 1 is very difficult to follow. It should include information about where randomization occurs, and when/now patients progress to next arms. I would assume there was a figure legend that got deleted in the process of manuscript formatting? Would have helped if there was at least a figure legend. 

For a such large study with multiple arms and randomization schemes, I assume there was considerable effort to ensure statistical power. Adding a paragraph or subsection about the statistical consideration can significantly strengthen the manuscript. 

Minor comments:

Background: The authors do mention this later, but I would elaborate more about FOXO1 fusions when discussing histological subtypes of RMS, and predictive biomarkers. The field is rapidly shifting to calling RMS subtypes as Fusion positive or Fusion negative RMS, and it would be helpful to elaborate on this prior to page 14, section 4.4. 

In page 5, I would consider formatting this section with a subtitle such as “Eligibility criteria and risk stratification”, then would follow by explaining treatment approaches and radomizations according to risk groups.  

Frontline treatment: On page 7 rows 244-248, the authors comment on ARST0531 and VAC/VI. While it is true that VAC/VI seemed to be equivalent to VAC, the VAC that was used in ARST0531 reduced the cyclophosphamide dose intensity, and led to poor outcome in ARST0531 compared to VAC, as reported in PMID 31174239.  As a result, VAC/VI is only used in the setting of trials such as ARST1431, and not as standard of care off study. I understand that this is a delicate topic, considering ARST1431 (and this study) will still use VAC/VI in the hopes of adding maintenance will overcome the inferiority of VAC/VI. So I might just adjust this paragraph to not explicitly state that VAC/VI is an alternative standard of care.  

Metastatic RMS: On page 8, the authors should add reference to PMID 26503200, which demonstrates that there are subgroups of metastatic RMS that have significantly better survival with intensification.

Comments on the Quality of English Language

Please proofread the manuscript again. Numerous simple grammatical errors that are probably carryovers from previous editions or edits. 

Author Response

Thank you

Round 2

Reviewer 2 Report

Comments and Suggestions for Authors

No additional concerns.